# An Artificial Intelligence Approach to Bloodstream Infections Prediction

**DOI:** 10.3390/jcm10132901

**Published:** 2021-06-29

**Authors:** Kai-Chih Pai, Min-Shian Wang, Yun-Feng Chen, Chien-Hao Tseng, Po-Yu Liu, Lun-Chi Chen, Ruey-Kai Sheu, Chieh-Liang Wu

**Affiliations:** 1College of Engineering, Tunghai University, Taichung City 407224, Taiwan; kcpai@thu.edu.tw (K.-C.P.); lunchi@thu.edu.tw (L.-C.C.); 2Department of Critical Care Medicine, Taichung Veterans General Hospital, Taichung City 40705, Taiwan; minnshyan@vghtc.gov.tw; 3Center for Infection Control, Taichung Veterans General Hospital, Taichung City 40705, Taiwan; funnytako@vghtc.gov.tw; 4Department of Infectious Diseases, Taichung Veterans General Hospital, Taichung City 40705, Taiwan; tedi3tedi3@hotmail.com (C.-H.T.); liupoyu@gmail.com (P.-Y.L.); 5Department of Computer Science, Tunghai University, Taichung City 407224, Taiwan; rickysheu@thu.edu.tw

**Keywords:** bloodstream infection, artificial intelligence, machine learning, model interpretation

## Abstract

This study aimed to develop an early prediction model for identifying patients with bloodstream infections. The data resource was taken from 2015 to 2019 at Taichung Veterans General Hospital, and a total of 1647 bloodstream infection episodes and 3552 non-bloodstream infection episodes in the intensive care unit (ICU) were included in the model development and evaluation. During the data analysis, 30 clinical variables were selected, including patients’ basic characteristics, vital signs, laboratory data, and clinical information. Five machine learning algorithms were applied to examine the prediction model performance. The findings indicated that the area under the receiver operating characteristic curve (AUROC) of the prediction performance of the XGBoost model was 0.825 for the validation dataset and 0.821 for the testing dataset. The random forest model also presented higher values for the AUROC on the validation dataset and testing dataset, which were 0.855 and 0.851, respectively. The tree-based ensemble learning model enabled high detection ability for patients with bloodstream infections in the ICU. Additionally, the analysis of importance of features revealed that alkaline phosphatase (ALKP) and the period of the central venous catheter are the most important predictors for bloodstream infections. We further explored the relationship between features and the risk of bloodstream infection by using the Shapley Additive exPlanations (SHAP) visualized method. The results showed that a higher prothrombin time is more prominent in a bloodstream infection. Additionally, the impact of a lower platelet count and albumin was more prominent in a bloodstream infection. Our results provide additional clinical information for cut-off laboratory values to assist clinical decision-making in bloodstream infection diagnostics.

## 1. Introduction

Bloodstream infections (BSIs) are one of the leading causes of death. Patients diagnosed with BSIs have high morbidity worldwide, with an estimated overall crude mortality rate of 15–30% [1]. Early recognition and initiation of treatment is the key to successful treatment of bloodstream infection. In general, pathogens have been identified through blood culture, which is a time-consuming procedure due to the multiple steps required for identification [2]. Furthermore, delays in administering effective antibiotics could increase the risk of death [3]. Attempts have been made to develop effective biomarkers to detect BSIs. However, most laboratory-based methods fail in the early diagnosis of BSI [4,5].

Medical innovations powered by artificial intelligence are increasingly developing into clinically practical solutions. Machine learning or deep learning algorithms can effectively process the growing amount of data produced in various fields of medicine. Artificial intelligence can aid in the development of infection surveillance aimed at better recognizing risk factors, improving patient risk reduction, and detecting infections in a timely manner. Previous studies have developed different prediction models for bloodstream infections using various algorithms [6,7,8,9,10,11]. These studies collected data from different cohorts, such as general wards, intensive care units (ICUs), and surgical in-patients. Furthermore, different machine learning or deep learning algorithms were employed in these studies, with varying data collection windows. These studies examined the performance of a prediction model for BSI. Some studies presented excellent model performance [6,7], while others presented poor performance for identifying BSIs due to a data imbalance [8]. The Appendix A presents different prediction models, data cohorts, data collection windows, and model evaluations for the studies.

However, the data cohorts were different. There is scant evidence of AI and machine learning implementation in the field of BSI, and no consistent trend of effect has emerged, especially in the ICU. Additionally, fewer studies have explored model interpretation based on clinical features.

In this study, we aimed to develop an interpretable model to predict BSI in an Asian population. We used different approaches to evaluate which prediction performance of the model is better. Moreover, we tried to explain the relationship between clinical features and bloodstream infections using the Shapley Additive exPlanations (SHAP) visualized method.

## 2. Materials and Methods

### 2.1. Definition of Bloodstream Infection

We defined BSI as the growth of a clinically important pathogen in at least one blood culture. Contaminant microorganisms were classified as negative under the Clinical and Laboratory Standards Institute guidelines. The Appendix A presents the distribution of pathogens among the BSIs.

### 2.2. Data Acquisition

In this retrospective study, 4275 patients who were admitted to the Taichung Veterans General Hospital ICU were included. Between August 2015 and December 2019, 12,090 blood culture episodes were collected from these patients, with a total of 1680 BSI episodes and 10,410 non-BSI episodes. We found that most of the patients had two blood culture sampling episodes at the same time or the interval between two individual blood culture sampling episodes was less than 24 h. To avoid data sample noise, these episodes were randomly selected from only one episode for the data analysis. Additionally, we removed episodes in which the proportions of missing data for the clinical characteristics were more than 40%.

Finally, a total of 1478 bloodstream infections and 3597 non-bloodstream infections from blood culture tests were analyzed in our study. Figure 1 presents the flowchart of the study population selection. Table 1 reports the main characteristics of the overall population, and the Appendix A reports the patient characteristics in the training, validation, and test sets. The results of the *t*-test and analysis of variance (ANOVA) summary table for these data indicate that there were no statistically significant main or interaction effects.

### 2.3. Data Outcome and Prediction Window

The main objective of this study was to develop the early prediction of BSIs as a binary classification task. The prediction targets or the primary outcome of this study assessed bloodstream infections within a patient’s stay at the ICU. Figure 2 presents the BSI prediction task using 72 h of data to forecast the one set of blood culture tests after 24 h, which we designed as a 24-h prediction window after the feature window of 72 h. All vital signs data were collected between 96–24 h before blood culture was measured. The laboratory data were collected within one week to 24 h prior to the blood culture test because the data were measured infrequently. We used the mean of vital signs and laboratory tests as the feature values. Moreover, previous studies have indicated that the use of central venous catheters (CVC) increases the risk of BSIs. Therefore, the present study calculated the time of using CVC from ICU admission to 24 h prior to the blood test as a predictive feature. We also analyzed the period from ICU admission to 24 h prior to the blood culture measure in terms of the number of days.

### 2.4. Clinical Features Selection

This study is a retrospective analysis of the clinical data. The feature selection was reviewed based on the diagnostic criteria for sepsis and the risk factors of BSIs in the ICU [12,13]. Considering the available data from our electronic health record and the opinions of our expert domain, we collected thirty-two clinical variables as predictors of bloodstream infections, including patients’ basic characteristics, vital signs, laboratory data, and clinical information.

Vital signs were recorded every two hours in the ICU, including body temperature, respiratory rate, pulse rate, oximetry, systolic blood pressure (SBP), diastolic blood pressure (DBP), and Glasgow Coma Scale (GCS). Seventeen laboratory features were measured based on patients’ condition. Moreover, the usage time of central venous catheters (CVC), mechanical ventilation via endotracheal tube (ENDO), and Foley catheters were also included. Finally, we included the stay time of the patients’ ICU admission prior to the blood culture test. The feature characteristics are presented in Table 2. The difference between BSIs and non-BSIs was measured using a Student’s *t*-test for continuous variables. Additionally, a logistic regression analysis for crude and adjusted odds’ ratios is reported in Appendix A.

### 2.5. Study Design and Model Training

Figure 3 presents an overview of the predictive model’s established procedures, including the data pre-processing, model training, and model evaluation. Data pre-processing is an important issue in data analytics, including the removal of the outliers, missing value imputation, and data transformation. Moreover, some variables should be transformed because of different units, such as the white blood cell (WBC) count. For instance, the WBC count could be converted from 10^3/uL to K/uL. For removal of the outliers, we visualized all variables by using boxplots and discussed them with clinicians to identify the outliers, especially in vital sign records. We considered the plausible value as the inclusion criteria for vital signs by clinical expertise. The Appendix A presents the vital sign plausible values. The vital sign values that did not fall within the specific range were treated as outliers and excluded. For missing pre-processing values, we found that some lab tests, such as C-reactive protein (CRP) and glucose, were missing over 40% of their values/information (see Appendix A). The lab tests were not included in the final data analysis. We input the missing data by calculating the mean of the non-missing values in each column. Finally, a total of thirty features were used as predictors of BSIs.

For model development, the dataset was divided into a training set (60%), validation set (20%), and testing set (20%) after completing the data pre-processing. The validation dataset was evaluated to determine the model fit for the training dataset when tuning the hyperparameters and data preparation. The testing dataset was used to provide an unbiased evaluation of the final model fit on the training dataset [14]. To help examine the performance of the model, each model was evaluated by sensitivity, specificity, and the area under the receiver operating characteristic (AUROC) curve. An AUROC of 0.7 to 0.8 is considered acceptable, 0.8 to 0.9 is considered excellent, and greater than 0.9 is considered outstanding [15].

### 2.6. Data Analysis

The present study used conventional statistical approaches to analyze the data cohorts. For continuous variables, the difference between positive results and negative results by using an independent *t*-test were examined. For model training, five machine learning algorithms were used in this study, including logistic regression (LR), support vector machine (SVM), multi-layer perceptron (MLP), random forest (RF), and eXtreme Gradient Boosting (XGBoost). The logistic regression model was chosen as the representative linear model [16]. The SVM was chosen as the representative non-probabilistic binary linear classifier [17]. The MLP was based on an artificial neural network [18]. The RF and XGBoost models were chosen as representative ensemble learning and tree-based methods [19,20]. 

The main objective of this study was to develop an early prediction model of BSIs that could correctly identify positive BSIs. The predictive output of the machine learning model is represented as a probability, which should convert the value to the target class so that different threshold settings will perform and identify different numbers of BSI classes. The present study compared the different thresholds to examine the model prediction performance and attempted to find a trade-off between the classification of a BSI and non-BSI.

Additionally, we explored the features’ importance in the proposed prediction model. The ensembles of decision tree methods, such as RF and XGBoost, can provide estimates of feature importance from a trained predictive model based on Gini importance. Some studies have explored interpretable machine learning by using SHAP, a game theoretic approach, to explain the output of any machine learning model [21]. Furthermore, the SHAP value plot could present the positive and negative relationships of the predictors with the target variable. In this paper, the SHAP method was used to explore the importance of clinical features and their relationship to BSI events for the XGBoost model.

## 3. Results

### 3.1. Evaluation of Different Models

The present study compared five different algorithms to evaluate which performance was suitable for our dataset. Table 3 and Figure 4 present the prediction performance of BSI. The predictive result of the machine learning algorithm represents a risk probability of BSI. The default value is set 0.5, which means if the threshold of the model exceeds 0.5, the model will determine that the patient has BSI. Comparing the sensitivity and specificity, the XGBoost showed the highest sensitivity on the validation and testing datasets (0.724 and 0.706, respectively). Additionally, the RF showed the highest specificity on the validation dataset and testing dataset (0.927 and 0.940, respectively). A lower sensitivity was found for the SVM, RF, and MLP models. The sensitivity of the validation and testing datasets were determined for SVM (0.578 and 0.566, respectively), RF (0.565 and 0.577, respectively), and MLP (0.494 and 0.406, respectively).

In terms of specificity, the RF model performed with the highest specificity, which was over 0.9 on the validation and testing datasets. The LR model performed with the lowest specificity, scoring 0.660 on the validation dataset and 0.644 on the testing dataset. 

The prediction performance was further assessed based on the AUROC from the validation data and testing data. According to the AUROC results, the RF model performed the highest AUROC in the validation and testing datasets (0.855 and 0.851, respectively). The XGBoost algorithm also performed a relatively high AUROC for the validating and testing datasets of 0.825 and 0.821, respectively. In contrast, the LR and MLP models had the lowest AUROC values in the test dataset (0.685, 0.667, respectively).

According to the results, the XGBoost and RF machine learning methods present better prediction performance of BSI. The AUROC results are over 0.8, which means that the model is considered excellent for predicting BSI. The Brier score is measured from the model fit; the lower the brier score, the better the performance of the model. The XGBoost and RF models yielded acceptable Brier scores. However, the LR model performed the lowest in predicting BSI. 

### 3.2. Evaluation of Different Cut-Off Thresholds

According to machine learning techniques, the predicted results were represented as a probability. This probability is a value that ranges from zero and one and represents the input that belongs to the target class, which means the value can be converted to a class. For binary classification, the default cut-off threshold value is 0.5. This means that if a model’s predicted results have a probability greater than 0.5, they predict a BSI. However, the default cut-off threshold may not have the best model prediction. When the threshold was changed, the results of sensitivity and specificity also changed. This allowed us to explore the trade-off between sensitivity and specificity.

The present study compared the different cut-off thresholds to evaluate the performance of the model in identifying BSIs. The purpose of the present study was to correctly identify patients with BSIs, so the focus was on finding the highest the proportion of positives that were correctly identified. The present study analyzed the different cut-off thresholds to determine the trade-off threshold for the BSI prediction model. According to the results of the model evaluation, the RF and XGBoost models had the best prediction performance for BSI. The two models further examined the evaluation of different cut-off thresholds.

Figure 5 shows the performance statistics for BSI event prediction in the testing dataset. Here, the x-axis represents the probability of identifying patients with BSI events, and the y-axis is the number of patients. We found that most of the patients with BSI events presented a higher predictive probability, and the patients with no BSI events indicated that most of the patients’ predictive probability is relatively low. In changing the threshold to 0.41, the sensitivity received a better score (80.8%) and acceptable specificity (67.0%) for the XGBoost model (Table 4). On the other hand, when the cut-off threshold was set to 0.4, it presented a higher sensitivity (85.8%) and acceptable specificity (69.9%) for the RF model. Moreover, a cut-off threshold set to 0.3 also presented a similar result (Table 4). According to the results, there could be a trade-off for the threshold based on sensitivity and specificity. This is particularly desirable if the experts want to identify patients with BSI events correctly by using both sensitivity and acceptable specificity. The results of other algorithms presented in the Appendix A.

### 3.3. Clinical Features Importance and Visualization

Regarding the interpretability of the machine learning model, SHAP values were used to visualize and explain how these features affect BSI events within the XGBoost model. The SHAP values can explain and explore the results of the machine learning model by using a theoretic game approach. The method provides an overview of important features and visualizes the values of each feature for every data point (sample) [18].

Figure 6a presents the feature importance as the strongest predictor to effect BSIs. For the top 20 important features, there are 2 patient characteristics, 4 vital sign features, 12 laboratory features, 2 types of catheters (CVC and Foley), and an ICU-stay up to 24 h prior to the performance of a blood culture test. The results revealed that alkaline phosphatase (ALKP) and the use time of the central venous catheter (TOTAL_CVC) were associated with a higher risk of BSI events. Moreover, prothrombin time (PT) and platelet (PLT) were the third and fourth most important features. Additionally, we found that Apache II score and age seemed to be important features in predicting bloodstream infection.

Figure 6b summarizes the SHAP value plot by combining feature importance with feature effects. The y-axis is defined by the feature and the x-axis is defined by the Shapley value. The plot describes the features’ overall influence on the model prediction. Each point in each feature represents an individual case, with colors ranging from blue (low feature value) to red (high feature value). The data points further to the right represent the features that contribute to the higher risk of BSI for a given individual case. The data points to the left represent the features that contribute to the lower risk of BSI. The vertical line in the middle represents no change in risk. We found that the data points (individual cases) with higher ALKP values had a higher risk of BSI. Furthermore, some points with lower ALKP values also had a higher risk of bloodstream infection. In terms of the total usage time of a central venous catheter (TOTAL_CVC), the results reveal that the longer the usage time of the central venous catheter, the higher risk of BSI. The PT also revealed similar results. However, in contrast to PT, the data points with a lower PLT had a high risk of BSI. Additionally, most of patients with higher Apache II scores were correlated with an increased risk of BSI.

Furthermore, the lab tests were used as the continuous variable in our prediction model. The present study examined the marginal effect of laboratory tests on the predicted outcome of a machine learning model using a SHAP dependence plot [22]. Figure 7 shows the dependence plot for PT, PLT, and albumin (ALB). The results showed that the value of prothrombin time over approximately 12.3 s were associated with a higher risk of BSI. Patients had a high risk of BSI when their PLT value was below approximately 120 K/uL. Consistent with the trend of PLT, ALB levels less than approximately 2.73 g/dL increased the risk of BSI. According to the dependence plot in Figure 7, we found cut-off values of laboratory features that provided additional clinical information to predict BSI for clinicians.

## 4. Discussion

In the present study, we used multiple machine learning algorithm approaches to develop an early prediction model for bloodstream infections. The prediction model achieved good performance in the validation dataset and testing dataset by using RF and XGBoost algorithms (AUROCs ranging from 0.821 to 0.855). The results demonstrated a good model fit for the tree-based ensemble methods. Compared to previous studies, the logistic regression model showed a range of AUROC values between 0.6 and 0.83 [23]. Lee et al. developed an early detection of bacteremia model using an artificial neural network approach. The AUROC results achieved 0.727 (95% CI, 0.713–0.727) and had a higher sensitivity (0.810) [6]. Ebrahim Mahmoud et al. also developed a prediction model for BSI among hospitalized patients [9]. However, these population studies were not conducted on critically ill patients. Roimi et al. developed an early diagnosis of BSI using machine learning for ICU patients. The study presented excellent AUROCs in two medical centers (0.89 ± 0.01 and 0.92 ± 0.02) [7]. 

We further identified the cut-off threshold for a trade-off between sensitivity and specificity. Some studies have compared the different cut-off thresholds to examine the model performance. In BSI predictive implementation, the evaluation of model performance focused on detecting patients with BSIs correctly [8]. The results showed that the trend of sensitivity and specificity of different cut-off thresholds was consistent, which is important for future research.

In terms of the features’ importance, the results showed that the ALKP laboratory test is the most important feature in predicting BSIs. Furthermore, it is consistent with a previous study. Lee et al. identified ALKP as one of the most influential features for BSI. We found that some patients with low ALKP were associated with high risk of BSI. It seems that there may be a sub-group of patients where low ALKP is actually a very important predictor of BSI, even if previous studies indicated that the high ALKP is positively associated with BSI. Further studies could consider the sensitivity analysis that excludes this sub-group of patients and explores the relationship between other clinical characteristics and BSI in this group. We also identified the total duration of CVC as an important risk factor, which has been observed in many studies [6,24]. Hence, our model effectively identified the risk factors associated with the development of BSIs. We also found that some laboratory features, such PT, PLT, and ALB, are important in the development of BSIs, as confirmed by other studies [6,9,11]. According to the analysis of the dependence plot of the laboratory features, we could observe the cut-off values of the features with higher risk of BSI. The results provided helpful laboratory tests information for identifying the risk of BSI. We also found that the length of an ICU stay until the blood culture test plays an important role in the development of BSIs. A retrospective cohort study using data from 113,893 admissions revealed an association between the length of a hospital stay and an increased risk of BSIs [25]. Other studies have also indicated that the hospital-to-blood culture period has a positive effect on BSIs [6,26]. Overall, most of the studies present similar results, even though they are sampled from different populations, such as the United States, Israel, Saudi Arabia, and South Korea. We discovered some consistent features regarding the important risks of BSIs in these studies. 

However, the present study has some limitations. First, the data were collected from a single medical center, and external validation is required, even though the independent validation process was implemented in our study. Second, the model was developed based on 72 h data and a 24 h prediction window, and patients who stayed in the ICU for less than 96 h were excluded. Third, the time-to-event features (CVC, Foley, and ENDO) were not evaluated using an alternative binary classification in our model; we did not compare the different types of features to find the best predictors. Lastly, the data was slightly imbalanced, which means the precision of the model training was relatively low because of the lower number of BSI events.

## 5. Conclusions

The present study developed a machine learning model for the early identification of patients with a high risk of BSIs in the ICU. The performance of the prediction was found to be compatible with previous studies. We explored how different cut-off thresholds affected the prediction performance. Moreover, we used the SHAP method to explain the results of the prediction model.

In general, our data highlight the importance of prediction models powered by artificial intelligence. Further studies are needed to validate this model through conventional clinical trials.

## Figures and Tables

**Figure 1 jcm-10-02901-f001:**
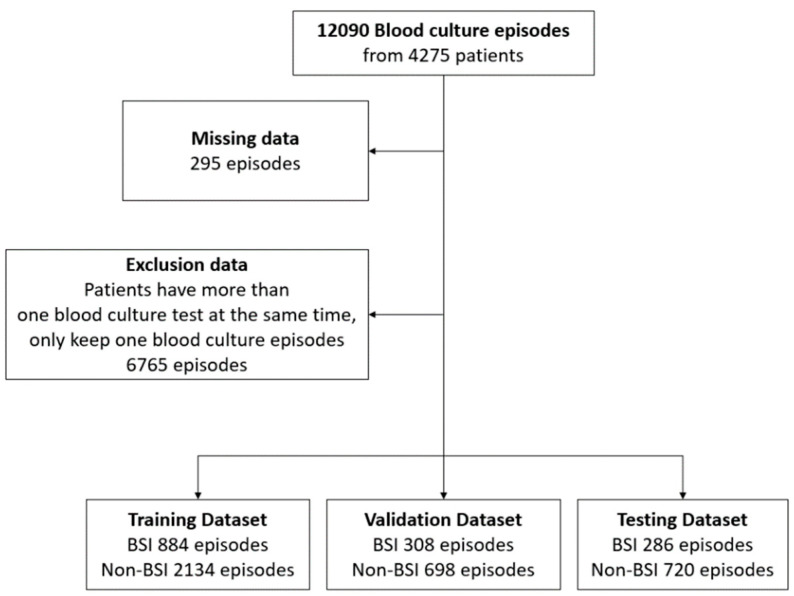
Data cohort workflow.

**Figure 2 jcm-10-02901-f002:**
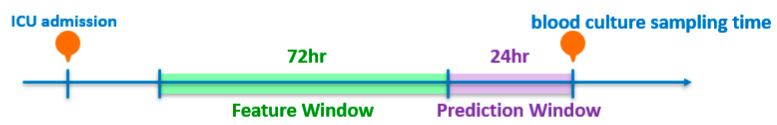
Representation of the BSI prediction task.

**Figure 3 jcm-10-02901-f003:**
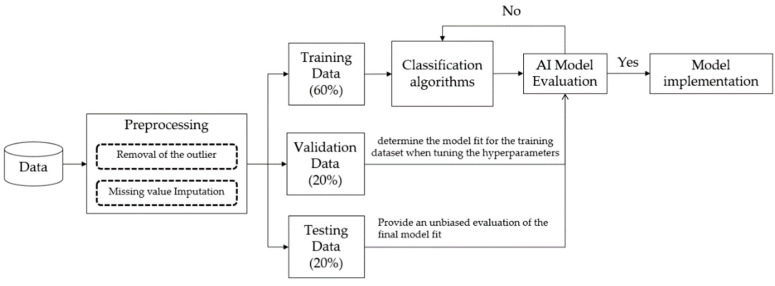
Overview of the model design.

**Figure 4 jcm-10-02901-f004:**
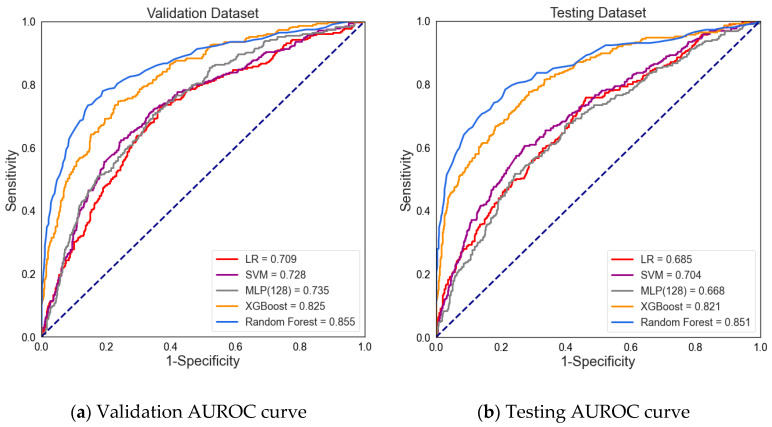
The area under the receiver operating characteristic (AUROC) curve for (**a**) the validation dataset (validation AUROC curve) and (**b**) testing dataset (testing AUROC curve).

**Figure 5 jcm-10-02901-f005:**
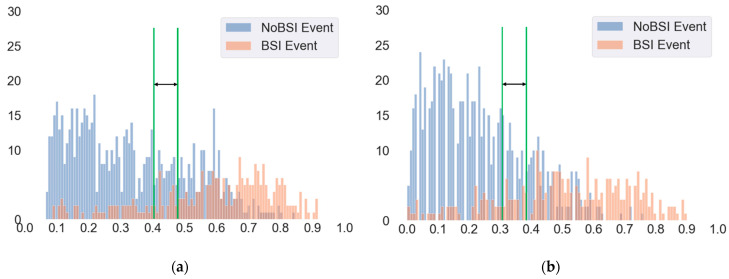
Model performance statistics for (**a**) the trade-off threshold for the XGBoost model and (**b**) the trade-off threshold for the RF model.

**Figure 6 jcm-10-02901-f006:**
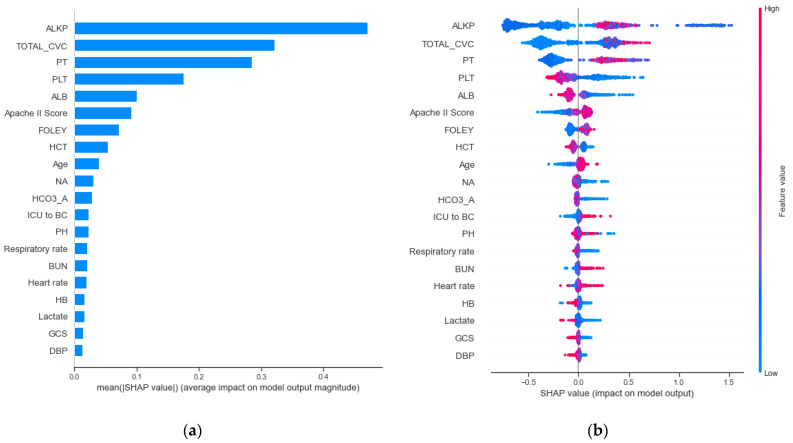
Model interpretation and visualization of for (**a**) feature importance and (**b**) the relationship between features and bloodstream infection.

**Figure 7 jcm-10-02901-f007:**
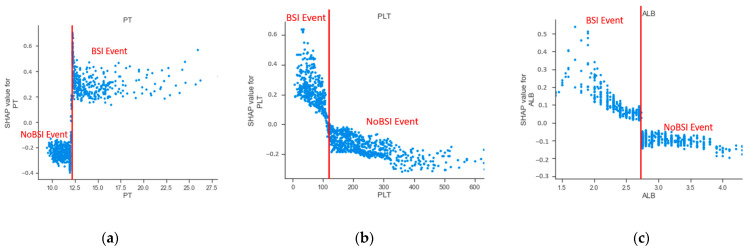
The SHAP independence plot of laboratory features for (**a**) prothrombin time, (**b**) platelet count, and (**c**) albumin.

**Table 1 jcm-10-02901-t001:** Patient demographics of the overall study population.

	All (*n* = 5030)	BSI (*n* = 1478)	Non-BSI (*n* = 3552)	*p*-Value
**Basic characteristics**				
Age, years	66.36 ± 15.83	67.50 ± 14.74	65.89 ± 16.23	<0.001
Sex (male)	3354 (65.15%)	1040 (70.37%)	2314 (66.68%)	<0.001
Charlson comorbidity index	2.23 ± 1.43	2.35 ± 1.41	2.19 ± 1.43	<0.001
APACHE II score	25.73 ± 6.13	26.47 ± 6.11	25.42 ± 6.12	<0.001
**Divisions**				<0.001
MICU	2024 (40.24%)	615 (41.61%)	1409 (39.67%)	
SICU	1292 (25.69%)	489 (33.09%)	803 (22.61%)	
CCU	727 (14.45%)	252 (17.05%)	475 (13.37%)	
*CV*	404 (8.03%)	115 (7.78%)	289 (8.14%)	
*CVS*	323 (6.42%)	137 (9.27%)	186 (5.24%)	
NICU	987 (19.62%)	122 (8.25%)	865 (24.35%)	
*NEURO*	166 (3.30%)	25 (1.69%)	141 (3.97%)	
*NS*	821 (16.32%)	97 (6.56%)	724 (20.38%)	
**The etiology for ICU admission**				<0.001
Scheduled surgery	241 (5.09%)	49 (3.49%)	192 (5.76%)	
Emergency surgery	113 (2.39%)	31 (2.20%)	82 (2.46%)	
NS surgery, scheduled	50 (1.06%)	4 (0.28%)	46 (1.38%)	
NS surgery, emergency	318 (6.71%)	22 (1.56%)	296 (8.89%)	
Acute respiratory failure	1130 (23.85%)	325 (23.12%)	805 (24.17%)	
Pneumonia	327 (6.90%)	92 (6.54%)	235 (7.05%)	
Sepsis, non-pneumonia	308 (6.50%)	125 (8.89%)	183 (5.49%)	
Acute cardiac conditions	544 (11.48%)	164 (11.66%)	380 (11.41%)	
Acute neurological conditions	170 (3.59%)	29 (2.06%)	141 (4.23%)	
Pulmonary embolism	0 (0%)	0 (0%)	0 (0%)	
Acute renal conditions	58 (1.22%)	14 (1.00%)	44 (1.32%)	
Acute GI condition	688 (14.52%)	276 (19.63%)	412 (12.37%)	
Post-PCI	22 (0.46%)	6 (0.43%)	16 (0.48%)	
OHCA/INCA	39 (0.82%)	12 (0.85%)	27 (0.81%)	
Others	729 (15.39%)	257 (18.28%)	472 (14.17%)	
**Outcomes**				
ICU stay, days	25.65 ± 20.05	34.10 ± 25.39	22.13 ± 16.09	<0.001
Hospital stay, days	49.12 ± 40.57	58.37 ± 39.95	45.27 ± 40.21	<0.001

**Table 2 jcm-10-02901-t002:** The clinical characteristics of the BSI and non-BSI groups.

Clinical Variable	BSI (*n* = 1478)	Non-BSI (*n* = 3552)	*p*-Value	Standard Cut-off
Vital sign				
Temperature (°C)	36.62 ± 0.49	36.66 ± 0.48	0.007	
SBP (mmHg)	121.40 ± 14.95	123.78 ± 15.31	<0.001	
DBP (mmHg)	66.46 ± 10.50	67.95 ± 10.79	<0.001	
GCS	7.37 ± 3.20	7.56 ± 3.53	0.065	
Heart rate (bpm)	93.36 ± 15.57	90.28 ± 15.58	<0.001	
Respiratory rate (breath/min)	19.03 ± 3.53	18.65 ± 3.61	<0.001	
Laboratory				
Albumin (g/dL)	2.64 ± 0.55	2.84 ± 0.59	<0.001	3.5–5
Alkaline phosphatase (U/L)	211.00 ± 181.11	191.23 ± 220.81	0.015	50–190
BUN (mg/dL)	53.81 ± 37.91	42.31 ± 33.90	<0.001	5–25
Creatinine (mg/dL)	2.26 ± 1.90	1.94 ± 1.99	<0.001	0.5–1.4
CRP (mg/dL)	11.09 ± 8.99	9.93 ± 9.80	0.016	<0.3
Glucose (mg/dL)	180.08 ± 91.35	182.99 ± 103.07	0.574	70–200
HCO3-A (mmol/L)	23.38 ± 5.16	24.40 ± 5.24	<0.001	22–26
Hematocrit (%)	27.42 ± 4.78	29.26 ± 5.79	<0.001	37–52
Hemoglobin (g/dL)	9.13 ± 1.46	9.76 ± 1.79	<0.001	12–17.5
Potassium(K) (mEq/L)	3.90 ± 0.67	3.95 ± 0.67	0.011	3.5–5.3
Na (mEq/L)	140.26 ± 7.41	140.76 ± 7.01	0.024	137–153
pH (blood gas)	7.43 ± 0.07	7.43 ± 0.07	0.83	7.35–7.45
Platelet count (/UL)	140.39 ± 107.39	196.45 ± 125.71	<0.001	150–400
PO2-A (mmHg)	122.80 ± 49.18	124.50 ± 62.55	0.414	80–100
Prothrombin time (PT) (s)	13.98 ± 6.68	12.77 ± 5.48	<0.001	9.5–11.7
WBC (/UL)	11.81 ± 7.55	12.22 ± 7.13	0.066	3.5–11
Lactate (mg/dL)	18.32 ± 17.01	16.08 ± 14.54	<0.001	3.0–12
Clinical information				
ICU day to blood culture, days	19.70 ± 17.53	8.99 ± 4.86	<0.001	
Central venous catheter (h	1233.57 ± 2965.12	698.78 ± 3261.02	<0.001	
ENDO (h)	2927.38 ± 5080.96	1600.29 ± 3972.20	<0.001	
FOLEY (h)	528.04 ± 536.07	325.87 ± 632.31	<0.001	

Abbreviations: BSI, bloodstream infections; Non-BSI, non-bloodstream infections; SBP, systolic blood pressure; DBP, diastolic blood pressure; GCS, Glasgow Coma Scale; BUN, blood urea nitrogen; CRP, C-reactive protein; WBC, white blood cell.

**Table 3 jcm-10-02901-t003:** Prediction performance of BSI using different algorithms.

Dataset	Algorithms ^1^	AUROC (95% CI)	Sensitivity (95% CI)	Specificity (95% CI)	Brier Score
Validation dataset	LR	0.709 (0.679–0.737)	0.679 (0.624–0.728)	0.660 (0.625–0.695)	0.218
	SVM	0.728 (0.699–0.756)	0.578 (0.522–0.632)	0.779 (0.747–0.809)	0.195
	MLP	0.735 (0.707–0.761)	0.494 (0.438–0.549)	0.832 (0.803–0.858)	0.231
	XGBoost	0.825 (0.802–0.849)	0.724 (0.672–0.771)	0.777 (0.744–0.806)	0.165
	RF	0.855 (0.832–0.877)	0.565 (0.509–0.619)	0.927 (0.905–0.944)	0.139
Testing dataset	LR	0.685 (0.653–0.715)	0.615 (0.558–0.670)	0.644 (0.609–0.679)	0.223
	SVM	0.704 (0.673–0.733)	0.566 (0.508–0.623)	0.756 (0.723–0.786)	0.199
	MLP	0.668 (0.633–0.698)	0.406 (0.350–0.463)	0.811 (0.781–0.838)	0.254
	XGBoost	0.821 (0.795–0.843)	0.706 (0.651–0.756)	0.775 (0.743–0.804)	0.163
	RF	0.851 (0.824–0.872)	0.577 (0.519–0.633)	0.940 (0.921–0.955)	0.134

^1^ LR: logistic regression, SVM: support vector machine, MLP: multi-layer perceptron, XGBoost: eXtreme Gradient Boosting, RF: random forest.

**Table 4 jcm-10-02901-t004:** The model performance of different cut-off thresholds in the test dataset.

Algorithms	Cut-Off Threshold	Sensitivity	Specificity	Precision	True Positive	True Negative	False Positive	False Negative
RF	0.3	82.9%	69.2%	51.6%	237 (23.6%)	498 (49.5%)	222 (22.1%)	49 (4.9%)
	0.4	69.9%	85.8%	66.2%	200 (19.9%)	618 (61.4%)	102 (10.1%)	86 (8.6%)
	0.41	68.2%	86.0%	65.9%	195 (19.4%)	619 (61.5%)	101 (10.0%)	91 (9.1%)
	0.5	57.7%	94.0%	79.3%	165 (16.4%)	677 (67.3%)	43 (4.3%)	121 (12.0%)
	0.53	51.0%	96.5%	85.4%	146 (14.5%)	695 (69.1%)	25 (2.5%)	140 (13.9%)
	0.6	38.8%	98.2%	89.5%	111 (11.0%)	707 (70.3%)	13 (1.3%)	175 (17.4%)
	0.7	21.3%	99.6%	95.3%	61 (6.1%)	717 (71.3%)	3 (0.3%)	225 (22.3%)
XGBoost	0.3	89.9%	48.8%	41.1%	257 (25.5%)	351 (34.9%)	369 (36.7%)	29 (2.9%)
	0.4	82.2%	64.3%	47.8%	235 (23.4%)	463 (46.0%)	257 (25.5%)	51 (5.1%)
	0.41	80.8%	67.0%	49.4%	231 (23.0%)	483 (48.0%)	237 (23.6%)	55 (5.4%)
	0.5	70.6%	77.5%	55.5%	202 (20.1%)	558 (55.5%)	162 (16.1%)	84 (8.3%)
	0.53	67.1%	81.0%	58.4%	192 (19.1%)	583 (58.0%)	137 (13.6%)	94 (9.3%)
	0.6	53.5%	90.7%	69.5%	153 (15.2%)	653 (64.9%)	67 (6.7%)	133 (13.2%)
	0.7	33.2%	97.9%	86.3%	60 (5.9%)	693 (68.9%)	5 (0.5%)	248 (24.7%)

XGBoost: eXtreme Gradient Boosting, RF: random forest.

## Data Availability

The data are not publicly available due to the constraint of the Taiwanese Personal Information Protection Act.

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
