# Peer review of "An Artificial Intelligence Approach to Bloodstream Infections Prediction"

_jcm, 2021, doi:10.3390/jcm10132901_

Round 1
Reviewer 1 Report
The authors attempted to develop a predictive tool for blood stream infections by using multiple advanced methods.
Overall, the methods are appropriate.
However, major improvements are needed for this manuscript starting with reviewing the English by an English native writer. The low level of language made it very difficult to clearly understand the text. Here are some suggestions and comments.
Abstract: review the language and add more details about the study population and methods. Please report more significant and relevant results.
Introduction: it would be relevant to report a review and description of main characteristics, methods, and results of other studies that attempted to develop a prediction tool on blood stream infections, in addition to these studies's major limitations. A supplementary table that summarizes these studies would be very helpful for the reader and would highlight the added value of this study.
Methods:
- The main characteristics of the overall population and in each dataset should be described (age, gender, comorbidities, reason for hospitalization and ICU admission, treatments, surgeries...etc). These characteristics should be compared between derivation and validation datasets to insure randomness of selection and absence of major biases.
- The choice of covariates included in the predictive modeling should be justified.
- Clinical characteristics should also be reported by categories (standard cut-offs) that are used for lab tests. Although continuous values are more relevant for predictive modelling, they are not relevant in clinical settings and for clinicians. Logistic regression and Random Forest methods could be conducted using continuous and categorial covariates.
- Major methodological limitation is the absence of adjustment to patients' characteristics such as age and comorbidities. The models should be re-conducted to adjust for these covariates at least under a sensitivity analysis.
- Evaluation of different cut-off threshold: the authors should clearly define what the threshold represents, what is the reported measure. Again, the manuscript should be revised by an English native speaker to improve the text.
Results:
- The population and datasets main characteristics should be reported. A Table 1 should be added to show these characteristics with clinical co-variates reported in frequencies according to standard lab tests.
- The section 3.1 should be summarize the most relevant results and avoid repeating all the results in Table 2.
- Table 2: please add confidence intervals for each reported measure (AUC, sensitivity, and specificity)
- Please report the results of the crude and adjusted logistic regression as this method is the most frequently used and familiar to clinicians. These results could be a supplementary table.
- Please report the results and graphs of the other algorithms when applicable.
- Please report the measure of models fitting when relevant.
- Please report the frequency of missing data that was imputed and outliers excluded.
- Table 3: please add the measure of each parameter (% and/or n) and report sensitivity, specificity, and precision in %. Please add the definition used to cut-off.
Discussion:
- Lines 225-226: it should be discussed how and why this infection would be different in the Asian population and how this difference was taken into account in this study and models.
Reviewer 2 Report
Pai et al. present a large study of multiple ML algorithms to detect bloodstream infection in ICU patients. It has notable strengths by virtue of its size and the various methods it evaluates, and in the performance of the ensemble algorithms. I especially support the use of distinct validation and testing sets. However, I think the study could benefit significantly from further methodological details, a stronger linking of the work to the clinical implications (including stronger descriptions of the underlying clinical and microbiological data), and tightening of the English syntax, to increase interpretability for patients and clinicians. My comments, in roughly chronological order, were the following:
- Lines 42-43 is a pretty broad statement and would benefit from some cites
- Line 52: Providing information in a supplement showing the breakdown of pathogens among the BSIs would be very helpful.
- Line 61-62: I do not understand what this means.
- Figure 1: How were the exclusion data picked? If a patient had more than one culture, how did you select which to retain? Great use of split evaluation sets.
- Lines 72-73: Please explain more about the prediction window. Do you mean after the 72 hours you would be predicting risk of BSI in the following 24 hours? Were the time-to-culture variables lagged by 24 hours to account for the prediction window interval? Also, many of the vitals and possibly labs are measured multiple times in 72 hours. How did you arrive at a single value for each patient – average? (See also Table 1)
- Lines 98-114: Need additional details here or in a supplement on this data pre-processing. This is actually a very important and subjective process, so transparency and reproducibility require more details around how you excluded outliers (what criteria when identifying?) and when you say you used the mean to impute missing data, was that just the raw mean for the whole sample, for only those with the same outcome status, or how? Also, was missingness very high for many of these variables?
- Figure 3: When you used the validation set to ensure there was not overfitting, was the model tuned at this stage? Just wanting to better understand.
- Line 144: How did you select variables for the logistic regression model, or did you simply use all variables?
- Table 2: Where were these single se and sp values obtained from? ROC curves are a continuum of various sensitivities and specificities, so were these the values at the “optimal” cutoff? If so, what criteria were used to identify that cutoff. Details in the figure legend or text would be helpful.
- Lines 182-183: I do not understand what these lines mean.
- Table 3: While the columns denoting false positive and false negative numbers are helpful, I think they highlight an important challenge, which is that the datasets were artificially balanced to improve classifier performance, but in the real-world, BSI is rare (<10% in their data). Hence, we know specificity matters more to overall performance when outcomes are rare, but the authors acknowledge that sensitivity is more important in this context. That means any algorithm that starts to try to prioritize sensitivity will have a very high number of false positives because it comes at the tradeoff of high specificity. This is a tension common to many applications of machine learning to rare outcomes in the health sector, and doesn’t mean the authors have done anything incorrect, but I think it warrants a footnote here or a line in the discussion section.
- Line 196: The use of SHAP methods was helpful and provided good illustration, but I’d encourage the authors to add an additional line or two further explaining what SHAP is. Many readers, especially clinicians, will not be familiar with this acronym.
- Discussion section: Most of the most important features involved a time-to-event component. Did the authors also evaluate an alternative binary classificatioin and let the models pick the best predictor(s)? If not, I’d consider adding this as a limitation with sentence or two, as variables with time components can have more complex interpretations that can be difficult to parse and can be skewed. For example, could you only follow duration of CVC for patients who were hospitalized prior to ICU admission and was this data missing for patients who are admitted from outside directly to ICU? I wonder how much the time component is simply a marker for patients with more complex or severe illnesses that have left them hospitalized for longer. This doesn’t mean the algorithms are wrong – prediction is prediction – but this may warrant a little fleshing out in the discussion if the authors are also choosing to focus on interpretability of the features themselves.
Round 2
Reviewer 2 Report
The authors have clearly put a substantial amount of work into the revisions and additional analyses, which improve the manuscript a lot. I only have a few small remaining comments, but one of them (asterisked) I think is essential to clarify further in the manuscript:
1)* I had confusion around this during the first review, and I am still confused even after the edits and response about the feature vs prediction window, and when features were collected. The diagram suggests that features are only collected up until 24 hours before blood culture (i.e, there is a 24-hour "lag" or censor window). However, if that is the case then isn't line 98 incorrect where it says features were collected 72 hours before blood culture? I believe the features were actually collected between 96-24 hours before blood culture, yes? Similarly, at lines 103-104, you discuss your time vairables such as time from admission to culture, as if the time components include the 24 hour prediction window (shouldn't it be time from admission to 24 hours prior to culture, since that is when the prediction window starts?). I have less concerns that this would inclusion of time right up until culture for the time-component variables would bias the analysis extensively, but for clarity around the methods I think this needs to be clarified for readers. I have more concerns about the lab values if they included features in the 24 hours up to culture time, as a patients clinical status could change very quickly if they deteriorate in that window that might skew associations.
Minor comments:
2) Line 137 and others: I would consider rephrasing 'experts domain'. I believe you mean something like subject-matter expert or clinical expertise?
3) If I am interpreting the SHAP plot correctly, this is really fascinating because it appears that for alkaline phosphatase, there is a long right tail where low alkp is associated with very high risk of BSI. This suggests that there could be heterogeneity of effects here or something else interesting going on which even a good RF algorithm wouldn't be able to fully accomodate probably in its branching. I'd consider mentioning this finding and the need for further study (if my interpretation is correct) at lines 316-318 in the discussion, e.g., that there may be a sub-group of patients for whom low ALKP is actually a very important predictor of BSI, even if in general it is high ALKP that is positively associated with BSI. This seems to be a real value of the SHAP method insofar as it lets you visualize these interesting findings. At the same time, I have to wonder, given the outsized effect of these patients on the model output, how much they may have compromised model performance, esp sensitivity. Not for this paper, which already has many analyses and supplementary tables, but at some point you might want to consider a sensitivity analysis that excludes these patients to see how model performance changes.
Author Response
Dear reviewer:
I am very grateful to your comments for the manuscript. According with your advice, we amended the relevant part in manuscript. Some of your questions and suggestions were answered. Please see the attachment.
